# Transient and Persistent Gastric Microbiome: Adherence of Bacteria in Gastric Cancer and Dyspeptic Patient Biopsies after Washing

**DOI:** 10.3390/jcm9061882

**Published:** 2020-06-16

**Authors:** Malene R. Spiegelhauer, Juozas Kupcinskas, Thor B. Johannesen, Mindaugas Urba, Jurgita Skieceviciene, Laimas Jonaitis, Tove H. Frandsen, Limas Kupcinskas, Kurt Fuursted, Leif P. Andersen

**Affiliations:** 1Department of Clinical Microbiology, Rigshospitalet, Henrik Harpestrengs Vej 4A, 2100 Copenhagen, Denmark; Tove.Havnhoj.Frandsen@rsyd.dk (T.H.F.); leif.percival.andersen@regionh.dk (L.P.A.); 2Department of Gastroenterology, Lithuanian University of Health Sciences, Eiveniu str. 2, LT-50009 Kaunas, Lithuania; juozas.kupcinskas@lsmuni.lt (J.K.); mindaugas.urba@lsmuni.lt (M.U.); laimas.jonaitis@lsmuni.lt (L.J.); limas.kupcinskas@lsmuni.lt (L.K.); 3Institute for Digestive Research, Lithuanian University of Health Sciences, Eiveniu str. 2, LT-50009 Kaunas, Lithuania; jurgita.skieceviciene@lsmuni.lt; 4Department of Clinical Microbiology and Infection Control, Statens Serum Institute, Artillerivej 5, 2300 Copenhagen, Denmark; THEJ@ssi.dk (T.B.J.); kfu@ssi.dk (K.F.)

**Keywords:** gastric microbiota, transient, persistent, culture, microbiome, sequencing, *Helicobacter pylori*

## Abstract

*Helicobacter pylori* is a common colonizer of the human stomach, and long-term colonization has been related to development of atrophic gastritis, peptic ulcers and gastric cancer. The increased gastric pH caused by *H. pylori* colonization, treatment with antibiotics or proton pump inhibitors (PPI) may allow growth of other bacteria. Previous studies have detected non-*Helicobacter* bacteria in stomach biopsies, but no conclusion has been made of whether these represent a transient contamination or a persistent microbiota. The aim of this study was to evaluate the transient and persistent bacterial communities of gastric biopsies. The washed or unwashed gastric biopsies were investigated by cultivation and microbiota analysis (16S rRNA gene-targeted amplicon sequencing) for the distribution of *H. pylori* and other non-*Helicobacter* bacteria. The number of cultured non-*Helicobacter* bacteria decreased in the washed biopsies, suggesting that they might be a transient contamination. No significant differences in the bacterial diversity were observed in the microbiome analysis between unwashed and washed biopsies. However, the bacterial diversity in biopsies shown *H. pylori*-positive and *H. pylori*-negative were significantly different, implying that *H. pylori* is the major modulator of the gastric microbiome. Further large-scale studies are required to investigate the transient and persistent gastric microbiota.

## 1. Introduction

### 1.1. *Helicobacter Pylori* Colonization of the Stomach

*Helicobacter* is a group of Gram-negative, curved or spiral-shaped bacteria, of which *Helicobacter pylori* is the most commonly known species [1]. *H. pylori* is able to colonize the human stomach by moving through the gastric mucin layer, and long-term colonization may increase the risk for development of atrophic gastritis, peptic ulcers and ultimately gastric cancer [2,3,4,5]. Approximately 30% of adults in developed countries and 80% of adults in developing countries are colonized with *H. pylori*, and of these, 1–3% further develop gastric cancer [6,7].

*H. pylori* has only been shown as a natural colonizer in humans, but another *Helicobacter* species capable of gastric colonization in humans is *Helicobacter heilmanii* [8,9]. The prevalence of infection with *H. heilmanii* is much lower than of *H. pylori* and it has been described less than 0.5% of patients undergoing upper gastric endoscopy [10].

Due to gastric peristalsis, mucus thickness, low pH and secretion of bile and acid, it was initially believed that no bacteria could survive in the inhospitable stomach environment [2,3,11]. Later, studies detected *H. pylori* as the only bacterium in stomachs with a healthy low pH, while more bacteria were detected at higher pH [12]. The mucus thickness and viscosity are pH-dependent, and a decreased acidity caused by *H. pylori* colonization, atrophic gastritis, treatment with antibiotics or proton pump inhibitors (PPI) may reduce the acidic protection of the stomach, leading to bacterial overgrowth and a higher diversity [13,14,15,16]. An alteration of the gastric bacterial composition has also been reported in cases of gastric cancer, with an increase of both oral and intestinal bacterial groups [16]. Recent studies have described the presence of a non-*Helicobacter* gastric microbiota, suggesting that other bacteria are able to live in the strict environment of the stomach [13].

### 1.2. Previous Studies of the Gastric Microbiota

Initial studies on gastric bacterial communities were reported in the 1980s [17]. Since then, several studies have investigated the presence of *H. pylori* and non-*Helicobacter* bacteria by cultivation methods and DNA-based methods [16,17]. Zilberstein et al. cultured gastric biopsies in aerobic and anaerobic conditions and found *Veillonella, Lactobacillus* and *Clostridium* to be the predominant bacterial groups [18]. They concluded that species of these bacterial groups may be transiently present [18]. Li et al. performed 16S rRNA gene amplicon sequencing on gastric biopsies from healthy people and patients with gastritis and observed a community dominated by species of *Prevotella, Streptococcus, Veillonella, Rothia* and *Haemophilus* [14]. Similar results were obtained by Bik et al. by 16S rRNA gene amplicon sequencing, where the dominating genera were found to be *Streptococcus, Prevotella, Rothia, Fusobacterium* and *Veillonella* [2]. A study by Delgado et al. on healthy patients identified *Lactobacillus* as one of the most abundant genera in the stomach as well as *Streptococcus* and *Propionibacterium* by 16S rRNA gene amplicon pyrosequencing [19]. A study by Dicksved et al. detected a bacterial community of *Streptococcus*, *Lactobacillus*, *Veillonella* and *Prevotella*, with a low abundance of *Helicobacter* [20]. The study found no significant differences in the gastric bacterial communities of patients with gastric cancer or dyspepsia [8]. Maldonado-Contreras et al. investigated the microbial composition by 16S rRNA gene amplicon hybridization and found non-*Helicobacter* bacteria such as *Proteobacteria, Firmicutes, Actinobacteria* and *Bacteroidetes* to be dominating [21]. Yu et al. identified similar phyla in the gastric stomach area and other body sites [22]. The authors performed a functional profiling of the stomach microbiota and concluded that though similar phyla were present in these areas, the microbiota of the different areas presented different functions [22]. A systematic review by Rajilic-Stojanovic et al. compared the results of papers investigating the gastric microbiota by next-generation sequencing (NGS) [16]. Approximately 2/3 of the described papers detected species of *Prevotella, Streptococcus, Veillonella, Neisseria, Fusobacterium* and *Haemophilus* and have described that the microbiota is subject-specific and differ between individuals [16]. More than 65% of the bacterial groups found in the stomach have also been identified in the human oral cavity, and many bacteria identified in the stomach may originate from the oral cavity or as reflux from the intestine [3]. Previous studies have not been conclusive about whether the detected non-*Helicobacter* bacteria represent a transient contamination of the stomach or if they belong to the persistent gastric microbiota.

### 1.3. *H. pylori* and the Non-Helicobacter Microbial Community

The human stomach has been described to contain a core microbiome mainly consisting of *Prevotella, Streptococcus, Veillonella, Rothia and Haemophilus* spp., influenced by diet, inflammation and medication [3]. However, the effect of *H. pylori* colonization on the bacterial diversity has not been established yet. Andersson et al. investigated the bacterial composition with 16S rRNA gene amplicon pyrosequencing and found complex microbial communities with a great diversity in the absence of *H. pylori* [23]. When present, *H. pylori* accounted for 90% of the bacteria, and diversity was decreased [23]. This agrees with other findings of a large variation in the bacterial community depending on the presence or absence of *H. pylori* [4,12,16,21]. The study by Maldonado-Contreras associated *H. pylori* colonization with an increase of *Proteobacteria*, *Spirochetes* and *Acidobacteria* and decrease of *Actinobacteria*, *Bacteroidetes* and *Firmicutes* [21]. Other studies have opposed the suggestion that the diversity of bacteria in the stomach is negatively affected by the presence of *H. pylori* [2,22,24]. As such, the significance of *H. pylori* on the presence of other bacteria in the gastric environment remains unknown. In a study by Sanduleanu et al., the non-*Helicobacter* bacteria in the stomach were found to contaminate both the gastric juices and the mucosa during gastric acid inhibition [15]. A study by Li et al. investigated the effect of washing on the bacterial content of gastric biopsies [14]. The majority of bacteria remained in the biopsies even after several washing steps, and in particular, *Streptococcus* were not removed [14]. Colonization with *H. pylori* has been associated with increased inflammation and development of gastric atrophy, which may cause overgrowth of other bacteria as the environment turns less acidic [25]. The presence of a complex microbial community in an atrophic stomach may further promote inflammation, malignancies and cancer development [25].

### 1.4. Microbial Effects on the Development of Gastric Cancer

*H. pylori* is classified as a class I carcinogen, and colonization with this species is associated with an increased risk for gastric cancer [26]. The precise mechanism for this development is not known in detail, but it may be affected by diet and interactions with the gastric microbiota [27]. It has been hypothesized that the presence of a specific microbiota promotes inflammation and that other bacteria in addition to *H. pylori* may further promote cancer development [26]. The pH is usually increased in the stomachs of gastric cancer patients, resulting in atrophic gastritis and subsequent loss of *H. pylori,* as the environment becomes less acidic [28]. Several studies of the gastric microbiota in the settings of gastric cancer have been performed. A study by Dicksved et al. compared the microbiota of biopsies from patients with gastric cancer and dyspepsia by terminal restriction fragment length polymorphism [20]. They identified complex compositions of bacteria in gastric cancer biopsies but with no significant differences in the microbial compositions compared to dyspepsia patient biopsies [20]. A low abundance of *H. pylori* and a dominance of species from *Streptococcus, Lactobacillus, Veillonella* and *Prevotella* were observed in biopsies from gastric cancer patients [20]. Another study by Ferreira et al. investigated the bacterial composition in biopsies from patients with atrophic gastritis and gastric cancer [29]. They detected a reduced microbial diversity in gastric cancer patients and found an abundance of less than 5% *H. pylori* in most gastric cancer patients [29]. Based on calculations of a microbial dysbiosis index, the presence of a dysbiotic microbiota in the gastric cancer patients was suggested, compared to the microbiota of individuals with atrophic gastritis [29]. In particular, the microbiota of gastric cancer biopsies showed increased abundances of the groups *Actinobacteria* and *Firmicutes*, while the presence of the groups *Bacteroidetes* and *Fusobacteria* were decreased [29]. Yu et al. analyzed the microbiota of gastric cancer biopsies and identified *H. pylori* as the most abundant bacterial species present, followed by bacteria associated with the oral environment [22]. A decrease in *Proteobacteria* and increase in *Bacteroidetes, Firmicutes* and *Spirochetes* were observed in gastric cancer tissue, but comparisons of microbiota in the corpus and antrum areas of the stomach showed no differences [22]. An altered bacterial composition has also been reported in cases of gastric cancer, with an increase of both oral and intestinal bacterial groups [16].

### 1.5. Definition of Persistent and Transient Microbiota and Microbiome

In this study, the gastric microbiota is defined as the persistent bacteria that adhere to the gastric mucosa. The gastric microbiome is defined as the collection of microbial genes in the stomach, which also include the community of non-adherent, transient bacteria [30]. Previous studies have not been able to agree on whether the bacteria identified in the stomach represent a true microbiota or are contamination from the oral cavity. It is therefore not clear which bacteria belong to the true gastric microbiota.

The main aim of this study was to investigate the presence of persistent and transient bacterial communities by comparing the changes in bacterial composition of washed and unwashed gastric biopsies. It is the first comprehensive attempt to distinguish between the transient and resident bacteria of the stomach using 16S rRNA gene sequencing.

The second aim of this study was to investigate and compare the gastric microbiota of gastric cancer patients and dyspepsia patients. The results will contribute to the ongoing debate about whether the process of gastric carcinogenesis is mediated through a microbial shift caused by *H. pylori*. Biopsies from both cancerous and non-cancerous tissue were included, and dyspepsia patients were involved as controls of a healthy microbiota. 

The hypothesis of our experimental setup is that *H. pylori* is the only true gastric microbiota of the stomach and will be the predominant bacterium that is able to remain in the washed biopsies. Although other studies have shown the presence of non-*Helicobacter* bacteria in the stomach, these will be expected to be present at least in the unwashed biopsies, as transient contamination from other niches [3].

## 2. Experimental Section/Materials and Methods

### 2.1. Sampling of Gastric Biopsies

Twenty-two patients with dyspepsia and twelve patients with gastric cancer were included in the study. The distribution was 68% female (*n* = 23 patients), 32% male (*n* = 11 patients) and an age range of 22–91 years (median 53 years). Clinical information about the patients is listed in Table 1. The exclusion criteria for the patients were age below 18 years, a previous history of *H. pylori* eradication, use of PPI within the last 3 months, use of antibiotics within the last 3 months, and previous treatment of gastric cancer. Three antral biopsies were taken about 4 cm from pylorus from patients with dyspepsia (*n* = 22 patients) by gastroscopy. From patients with gastric cancer (*n* = 12 patients), three antrum biopsies were taken about 4 cm from pylorus, and three biopsies were taken from the cancer area in the corpus. The biopsies were obtained between November 2017 and June 2019. The first biopsy was immediately used for histology to examine the presence of *H. pylori*. Histological slides were stained with haematoxylin/eosin, and a Giemsa stain was used to confirm the presence or absence of *H. pylori*. Figure 1 shows a histological sample positive for *H. pylori* infection.

All patients participating in the study have signed an informed consent form. The study protocol has been approved by Kaunas Regional Bioethics Committee (Protocol No: BE-2-10; P1-BE-2-31). 

The second biopsy was immediately transferred to the transport medium Portagerm pylori (bioMérieux, Marcy L’Etoile, France) and stored at −80 °C. The third biopsy was placed in a tube with sterile 4 °C PBS, collected by sterile forceps and washed for 15 s in 4 °C PBS and transferred to a new sterile tube containing 4 °C PBS. This step was repeated twice, after which the biopsy was placed in the transport medium Portagerm pylori (bioMérieux, Marcy L’Etoile, France) and stored at −80 °C. These biopsies were then transported to the Copenhagen University Hospital (Rigshospitalet) for culture.

### 2.2. Culture of Gastric Biopsies and Identification of Single Colonies

Biopsies were cultured on 7% defibrinated horse blood agar plates (SSI Diagnostica A/S, Hillerød, Denmark), placed in serum bouillon with 10% glycerol (SSI Diagnostica, Hillerød, Denmark), and frozen at −80 °C. The agar plates were incubated 6 days at 37 °C in microaerobic conditions (10% CO_2_, 5% O_2_); the observed growth was noted, and single colonies of each morphology were isolated on new 5% horse blood agar plates (SSI Diagnostica A/S, Hillerød, Denmark). The plates with the isolated cultures were incubated at 37 °C in microaerobic conditions until visible growth was seen (1–3 days). Each isolate was transferred to a MALDI-TOF target plate, treated with 1µl HCAA matrix (Bruker Daltonics, Billerica, MA, USA), analyzed with MALDI-TOF Mass Spectrometry (Bruker Daltonics, Billerica, MA, USA), and the protein profile was compared to the database Compass (Bruker Daltonics, Billerica, MA, USA) to identify the species. The biopsies were then transported to Statens Serum Institute for microbiome analysis (16S rRNA gene amplicon sequencing).

### 2.3. Microbiome Analysis (16S rRNA Gene Amplicon Sequencing)

DNA was extracted from the biopsies using a QIAamp DNA mini Kit (Qiagen, Hilden, Germany) according to the manufacturer’s instruction for tissues. For each batch of DNA extraction, a negative control without sample material was included for downstream analysis. DNA was amplified using a two-step PCR using a modified version of the published universal prokaryotic primers 341F (ACTCCTAYGGGRBGCASCAG) and 806R (ACTCCTAYGGGRBGCASCAG) targeting the V3-V4 regions of the 16S rRNA gene. Amplicons were sequenced on the Illumina MiSeq desktop sequencer (Illumina Inc., San Diego, CA, USA), using the V2 Reagent Kit.

#### 2.3.1. Library Preparation

Purified genomic DNA from each sample was initially amplified in a 25 µL reaction, using the REDExtract-N-Amp PCR ReadyMix (Sigma-Aldrich, St Louis, MO, USA) with 0.4 µM of each 16S rRNA gene primer and 2 µL template. The 16S PCR conditions were the following: an initial denaturation at 95 °C for 2 min, 20 cycles of 95 °C for 30 s, 60 °C for 1 min and 72 °C for 30 s and final elongation at 72 °C for 7 min. This PCR run is referred to as PCR1. The product from PCR1 was prepared for sequencing by a second PCR (referred to as PCR2), using the same PCR protocol as described above. PCR2 attached an adaptor A, an index i5 and a forward sequencing primer site (FSP) in the 5’ end of the amplicons and an adaptor B, an index i7 and a reverse sequencing primer site (RSP) to the 3’ end of the amplicons. DNA was quantified using the Quant-IT^TM^ dsDNA High Sensitive Assay Kit (Thermo Fisher Scientific, Waltham, MA, USA) and PCR2 products were pooled in equimolar amounts between samples. Agencourt AMPure XP bead (Beckman Coulter, Brea, CA, USA) purification was performed to remove undesirable DNA amplicons from the pooled amplicon library (PAL) in a two-step process. First, DNA fragments below 300 nucleotides length were removed by a PAL AMPure beads 10:24 ratio, following the manufacturer’s instructions, and eluted in 40 µL TE buffer (AM1). Secondly, large DNA fragments above 1kbp were removed by AM1 to AMPure beads 10:16 ratio as previously described. The resulting AMPure beads purified PAL (bPAL) was diluted to a final concentration of 11.5 pM DNA in a 0.001 N NaOH and used for sequencing on the Illumina MiSeq desktop sequencer (Illumina Inc., San Diego, CA, USA). The library was sequenced with the 500-cycle MiSeq Reagent Kit V2 in a 2 × 250 nt setup (Illumina Inc., San Diego, CA, USA). The sequencing was performed at Statens Serum Institute (SSI). 

#### 2.3.2. Bioinformatics

BION-META (http://box.com/bion), a newly developed analytical semi-commercial open-source package for 16S rRNA gene and other reference gene analysis classifying mostly species was used, and the data were processed following the previously described automated steps [31,32]. After BION-META analysis, taxonomy tables were made with the identified phylum, class, order, family, genus, species and number of reads. The data from the sequencing were submitted to European Nucleotide Archive with the accession number PRJEB38558.

#### 2.3.3. Statistics of Microbiome Analysis Results

The ten most abundant genera across all samples are shown in staggered bar plots in which samples are ordered according to a hierarchical clustering based on Bray Curtis dissimilarities and ward-linkage. The difference in the distribution of bacterial groups between biopsies was analyzed with an unpaired *t*-test using the graphing and statistics program GraphPad Prism version 5.01 (San Diego, CA, USA).

Analysis of the microbiome diversity was performed in the statistical computing program R, version 3.5.0 (R Foundation for Statistical Computing, Vienna, Austria) using the packages phyloseq v.1.24.3 and vegan v. 2.5-2. Figures were created using the packages ggplot2 v.3.2.0 and plotly v. 4.8.0. Alpha-diversity of samples as well as relative abundance of individual genera were compared pairwise between groups with Mann–Whitney rank sum tests and adjusted for multiple testing using Bonferroni correction. Principal coordinate analysis (PCoA) of samples was performed based BIONs species-level classification on Bray Curtis dissimilarity. Within-group similarities were compared to between-group similarities with analysis of similarities (ANOSIM test) using 1000 random permutations to estimate *p*-value.

## 3. Results

### 3.1. Comparison of Bacterial Composition in Unwashed Biopsies and Washed Biopsies

#### 3.1.1. A Decrease in Cultured Bacteria was Observed for the Washed Biopsies

The overall number of cultured non-*Helicobacter* species decreased in the washed biopsies compared to the unwashed biopsies, suggesting that many bacteria do not adhere to the tissue. A total of 27 biopsy pairs showed reduced or no growth in the washed biopsy compared to the growth observed in the unwashed biopsy (Table 2). Only 5 biopsy pairs showed unchanged growth, while 3 showed increased growth. 

The total number of colonies of cultured *Streptococcus* spp. decreased, but its relative abundance was higher in the washed biopsies compared to the unwashed biopsies (Figure 2, Table 3). The cultured bacteria in the biopsies were dominated by *Streptococcus* spp., followed by *Rothia* spp. and *Actinomyces* spp. (Table 3).

#### 3.1.2. Microbiome Analysis

Bacterial DNA was detected in all biopsies, even in those were no growth was observed. The ten most prevalent groups in the microbiome analysis were *Helicobacter* spp., *Streptococcus* spp., *Prevotella* spp., *Escherichia* spp., *Veillonella* spp., *Fusobacterium* spp., *Haemophilus* spp., *Rothia* spp., *Neisseria* spp., and *Alloprevotella* spp. (Figure 3). The average relative abundance of *H. pylori* increased in some of the washed biopsies compared to the unwashed, but this was not always observed in the individual samples (Table 4). The increase in *H. pylori* was therefore not significant. The “other bacteria” belong to over 100 different bacterial groups, of which some species were only present in few biopsies. The bacterial groups that were found in several of the biopsies are among the genera: *Abiotrophia, Aggregatibacter, Atopobium, Campylobacter, Capnocytophaga, Catonella, Corynebaccterium, Dialister, Eubacterium, Filifacter, Flavobacterium, Gemella, Granulicatella, Lachnoanaerobaculum, Lactobacillus, Leptotrichia, Megasphaera, Oribacterium, Parvimonas, Peptostreptococcus, Porphyromonas, Propionibacterium, Selenomonas, Solobacterium, Staphylococcus, Stenotrophomonas, Stomatobaculum* and *Treponema* (Table 5). The relative abundance of other non-*H. pylori* bacteria was similar between unwashed and washed samples, and none of the listed groups showed a significant change in relative abundance in either dyspepsia patients (Figure 3a) or gastric cancer patients (Figure 3b). No significant differences were observed in the bacterial distribution between the patient groups. Comparison of the bacterial diversity within the samples showed no significant differences between the unwashed and washed biopsies (Figure 3c), and similar bacterial species were clustered in both groups (Figure 3d). 

### 3.2. Comparison of Biopsies from Gastric Cancer Patients and Dyspepsia Patients

#### 3.2.1. Cultured Bacteria were Dominated by *Streptococcus* spp.

The biopsies from dyspepsia patients and gastric cancer patients were dominated by similar cultured bacteria (Figure 4). Species of *Lactobacillus* were cultured from several cancer patient biopsies but only from one dyspepsia patient biopsy (Table 2). The distribution of cultured bacteria in biopsies from gastric cancer patients and dyspepsia patients showed an increase in the relative abundance of *Streptococcus* and a decrease in *Actinomyces* spp. (Table 3, Figure 4).

#### 3.2.2. Microbiome Analysis Revealed Similar Distributions of Bacteria in Dyspepsia Patients and Gastric Cancer Patients

The average relative abundance of *H. pylori* was not significantly different in untreated biopsies untreated dyspepsia patients and gastric cancer patients (Figure 5a). For most bacterial groups, no significant difference was observed in the distribution of bacteria between dyspepsia patient and gastric cancer patient biopsies. However, a significant increase in the presence of *Prevotella* spp. was observed in dyspepsia patients (*p* = 0.0109) and of “other bacteria” in gastric cancer patients (*p* = 0.0349). This increase in other bacteria may be explained by the dominance of *Enterococcus* spp. in one biopsy pair, where more than 95% of the bacterial reads were identified as this. If this biopsy pair was removed from the data, the difference in “other bacteria” between the patient groups was not significant. The bacterial diversity in biopsies from dyspepsia patients and gastric cancer patients showed no significant differences within the groups (Figure 5b) or between the distribution of species in the groups (Figure 5c). 

A low percentage of the bacterial reads in biopsies from dyspepsia patients were identified as *Lactobacillus* spp. (<0.1% of total bacterial reads). Most biopsy pairs from gastric cancer patients (30 pairs) did not show presence of *Lactobacillus* spp.; 6 biopsy pairs showed a relative abundance of 0–0.5% of total bacterial reads, and 7 biopsy pairs showed a relative abundance of 0.5–2% *Lactobacillus* spp., while three biopsy pairs showed a high relative abundance of *Lactobacillus* spp., which both increased (from 11 to 27%) and decreased (from 23% to 0.4% and from 29% to 5%) between the unwashed and washed biopsies. This difference in *Lactobacillus* spp. abundance was not significant between dyspepsia and gastric cancer patients.

The biopsies from antrum and cancer area of gastric cancer patients showed no significant differences in the distribution of bacterial groups (Figure 6a, Table 6).

The diversity of bacteria in biopsies from antrum and corpus did not show differences in the bacterial diversity within the sample areas (Figure 6b) or between the two sample areas (Figure 6c). 

### 3.3. Presence of *H. pylori*

*H. pylori* was detected in 14 of 34 patients (41%) by histology. Culture of *H. pylori* was only successful for 4 biopsies from 2 patients, despite incubation for additional days. Microbiome analysis identified DNA from *H. pylori* in 16 of 34 patients (47%), and *H. pylori* was identified as the only species of *Helicobacter* in the biopsies. The difference between culture and 16S rRNA gene amplicon sequencing may be explained by the fastidious nature of *H. pylori*, which may be difficult to culture after storage. 

Three different distribution types of *H. pylori* and non-*Helicobacter* were observed; 4 biopsy pairs showed almost complete dominance of *H. pylori* (>90% of total bacterial reads in at least one of the biopsies) (Figure 7a), 16 biopsy pairs showed a mixed relative abundance of *H. pylori* and other bacteria (Figure 7b), and 26 biopsy pairs showed none or less than 1% of total bacterial reads identified as *H. pylori* (Figure 7c).

The biopsies determined positive or negative for *H. pylori* by histology showed significant differences in the bacterial diversity. The *H. pylori*-negative biopsies showed a significantly higher diversity than *H. pylori*-positive biopsies (*p*-value = 0.004353) (Figure 8a). In addition, a significant difference in the bacterial diversity was observed between biopsies determined positive or negative for *H. pylori* (*p*-value = 0.0009999), indicating that the presence of *H. pylori* may change the bacterial community to allow for a unique composition (Figure 8b).

## 4. Discussion

### 4.1. Washed Biopsies vs. Unwashed Biopsies

We observed a decreased bacterial growth from the washed biopsies compared to the unwashed biopsies in this study (Figure 2, Table 2). Based on our hypothesis, this might suggest that the bacteria removed by washing were not adhering to the biopsy and should not be considered a part of the gastric microbiota. *Streptococcus* spp. was the most dominant non-*Helicobacter* bacterial group in the biopsies, which may either be attributed to a higher starting concentration in the unwashed biopsy or an increased ability to adhere to the tissue, compared to the other bacteria in the biopsies.

The non-*Helicobacter* species identified in this study have previously been described as commensals of the oral cavity, upper airways and intestinal tract, including species of the dominant groups *Streptococcus, Rothia* and *Actinomyces* [11,16,20,21,25]. Intestinal bacteria such as *Enterococcus* spp. and *Escherichia* spp. were also identified with both culture and microbiome analysis and may be a sign of overgrowth from the intestines. Species of *Staphylococcus* were cultured from several biopsies, but the group was not among the 10 most common in the microbiome analysis, where between 0% and 4% of the reads were identified as *Staphylococcus* spp. The high detection in culture compared to microbiome analysis may be explained by selection of growth of *Staphylococcus*, as species of this group are facultative aerobic with a fast growth rate.

Our expectation was that only *H. pylori* would remain in the washed biopsies. This was not the case, as we observed the presence of many bacteria, of which the average relative abundance and diversity within the biopsies did not significantly change in the washed biopsies (Figure 3). This may suggest that some bacterial groups may remain in the biopsies and should be further investigated. 

### 4.2. Gastric Cancer Patients vs. Dyspepsia Patients

Comparison of the bacterial composition did not reveal significant differences in the bacterial diversity between biopsies from dyspepsia patients or gastric cancer patients. *Lactobacillus* was cultured, and DNA was detected in a higher abundance in biopsies from gastric cancer patients than in biopsies from dyspepsia patients, although not significantly. Development of gastric cancer has been described as a result of dysbiosis, and it is believed that a change in the gastric bacteria towards oral and intestinal bacteria may contribute to this process [19]. However, previous studies have not detected specific bacterial genera in only one of the patient groups [8,19]. It may be discussed if the presence of cancer cells causes alterations in the environment, which leads to changed microbial growth, or if the development is inversed. The increased presence of cultured *Lactobacillus* spp. in gastric cancer patient biopsies may also be an important difference between these patient groups, although the exact function is unknown. 

The expected result from this comparison was that a different bacterial composition would be observed in the cancer tissue compared to non-cancer patients, which was not confirmed by our results (Figure 6). To date, no studies have shown a connection between non-*Helicobacter* bacteria and development of pathologies such as gastric cancer in humans [11]. We observed an increased presence of *Lactobacillus* spp. in gastric cancer patient biopsies compared to dyspepsia patient biopsies in both the culture and microbiome analysis but cannot with certainty conclude that this bacterial group is involved in the development of gastric cancer. Four biopsies were positive for culture of *Lactobacillus* spp., where no reads were found in the microbiome analysis. However, 16 biopsies were negative for growth of *Lactobacillus* spp. but positive in the microbiome analysis. 16S rRNA-based analysis is more sensitive, and it is possible to detect the presence of bacteria in small quantities. An explanation for the difference in culture and microbiome results may be that 16S rRNA gene amplicon sequencing is not genus-specific, and as such, the 16S rRNA genes from several bacteria are amplified simultaneously. If *Lactobacillus* spp. are present in small quantities, the presence of a high amount of other DNA may drown it out, and it will not be detected. Combined with a selection in the culture for microaerobic-growing bacteria, the abundance of *Lactobacillus* spp. may appear higher in this experiment setting. This discrepancy between methods should also be considered in future culture- and sequencing-based studies. 

A study by Blaser and Atherton discussed whether *H. pylori* is a main driver of gastric cancer development, or if its presence is enough to change the stomach environment and allow for growth of other bacteria, which in turn increase the risk of gastric cancer [28]. The authors agree that long-term colonization causes inflammation and damage to the host tissues but that the mechanism leading to development of gastric cancer is not well defined [28]. In order to fully understand the pathological changes of other non-*Helicobacter* species in the stomach, infection assays should be performed on human cell cultures or model organisms with gastrointestinal tracts more similar to humans. 

### 4.3. Presence of H. pylori in the Biopsies

*H. pylori* was only cultured from the biopsies of two patients after extended culture, even though its presence in other patients was confirmed by histology and microbiome analysis. Further, microbiome analysis identified *H. pylori* in two biopsies, where the histology result was negative for *H. pylori*. This comparison of methods demonstrates that culture, histology and 16S rRNA gene sequencing are not always in accordance, and investigation of bacterial communities using several methods is preferred. During colonization or prolonged incubation, *H. pylori* take a coccoid form which may be difficult or impossible to culture in vitro [25]. Based on its slow growth and fastidious nature, culture of this species from clinical specimens is expected to be challenging. The culture results are still considered to be valid, as other species were cultivated, and thus, the medium and growth conditions must have been sufficient for bacterial growth. However, in order to culture all the present bacteria in the biopsies, several other medium types and atmospheres might have been considered. The relative distribution of *H. pylori* was expected to increase after wash, along with the removal of contaminating bacteria. This was not always observed (Figure 3). Studies have shown that it is possible to detect *H. pylori* with sensitive DNA-based methods in individuals previously shown negative for *H. pylori* with conventional methods [19]. This may also be the reason why a higher prevalence of *H. pylori* was observed with microbiome analysis compared to histology. The results from microbiome sequencing showed a variable distribution of *H. pylori* and non-*H. pylori* bacteria in the gastric environment. Three overall types of bacterial distribution were observed, suggesting that the stomach is more complex than previously thought (Figure 7). Natural variations in the microbiota caused by the geographical origin of the patients may also be of importance for this. The results from this study contribute to the agreement of a dynamic relationship between *H. pylori* and non-*Helicobacter* bacteria in the stomach, and previous discrepancies between studies may be caused by natural differences between patients. The ten most prevalent groups with microbiome analysis fits with the groups detected in other reported NGS-studies [19]. 

The biopsies showing positive or negative for *H. pylori* by histology analysis displayed significant differences in the bacterial diversity within and between the groups (Figure 8), suggesting that *H. pylori* may cause changes in the environment to allow for survival of other bacteria as suggested by Blaser and Atherton [28]. Our findings are also in agreement with the study by Maldonado-Contreras, which discovered a difference in the bacterial diversity based on *H. pylori*-status [21]. Patients negative for *H. pylori* presented a higher relative abundance of *Actinobacteria* and *Firmicutes*, whereas *H. pylori*-positive patients presented a higher abundance of *Proteobacteria* and *Acidobacteria*. 

The most common bacterial groups identified in the culture and microbiome analysis shows few similarities, and only two genera in the 10 most abundant groups from the microbiome analysis were cultured. However, both *Gemella* and *Granulicatella* were detected in several biopsies but not in high enough reads to be included in the top 10. The difference between the methods may be caused by selection of certain bacteria in the microaerobic incubation. Four of the 10 most common groups have been described as anaerobes or strict anaerobes (*Prevotella*, *Veillonella*, *Fusobacterium*, *Neisseria*), preventing them from growing in the microaerobic environment used in this study. *H. pylori* was detected in 47% of the biopsies by microbiome analysis but only cultured from two biopsy pairs. The biopsies were stored in the transport medium at −80 °C, which might have decreased the viability of *H. pylori.* It may be suggested that other bacteria identified by microbiome analysis were also present in the biopsies but were inhibited by the storage/freezing, and therefore unable to be cultured.

### 4.4. Limitations

It has been described that 80% of the bacteria that are identified with molecular methods in the human gut cannot successfully be cultured in vitro [16]. As we only cultured the biopsies in microaerobic environment, growth was selected for the microaerobically growing bacteria. The presence of these appeared higher than with microbiome sequencing. Several types of growth conditions and culture media would be required to select growth for all bacteria present in the biopsy. 

Culture-independent methods such as 16S rRNA gene amplicon sequencing or microbiome analysis may provide detailed information about the bacterial composition. One disadvantage may be that it does not differentiate between live and dead bacteria and between residents and contamination [2,10,19]. Other methods with the ability to distinguish between active and inactive bacteria, such as immunostaining or analysis of the metabolic activity, may also be considered for future investigations [22].

The results of this study are new as both culturing and sequencing are included as methods of detection, and the effect of washing or presence of *H. pylori* on the bacterial composition and diversity are investigated. Our results showed a decrease in the growth of some bacterial groups from washed biopsies, which are also known oral commensal bacteria. The species that remain in the tissue after wash must thus contain mechanisms for adhesion to avoid being removed. 

We present the first comprehensive paper attempting to distinguish between transient and resident bacteria in the stomach using a 16S rRNA gene amplicon sequencing approach and washing of biopsies. One other study has investigated the bacterial content of biopsies with a similar approach [14]. However, the study included only a small number of samples and used a taxon-specific quantitative PCR to define the taxa [14]. Future investigations in gastric microbiota should consider the presence of other bacteria in the stomach that may only be a transient contamination.

## 5. Conclusions

In conclusion, the number of cultured non-*Helicobacter* bacteria decreased in the washed biopsies, suggesting that they might be a transient contamination from oral cavity; however, in the microbiome analysis, no significant differences in the bacterial diversity were observed between unwashed and washed biopsies. The bacterial diversity in biopsies that were *H. pylori*-positive and *H. pylori*-negative was significantly different, implying that *H. pylori* is the major modulator of gastric microbiome. Further large-scale studies are required to investigate the transient and persistent gastric microbiota. This may include an increased number of samples, investigation of dysbiosis, a wider range of culture conditions and growth media, metatranscriptomic analysis, immunological assays or additional treatment of biopsies.

## Figures and Tables

**Figure 1 jcm-09-01882-f001:**
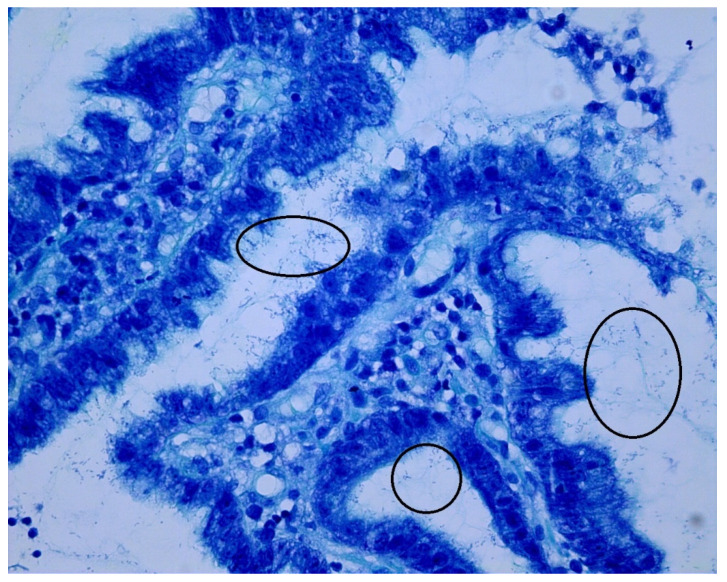
The positive results of *Helicobacter pylori* infection by Giemsa stain. Black circles indicate stained *H. pylori* (blue) that are attached to the gastric epithelial cells.

**Figure 2 jcm-09-01882-f002:**
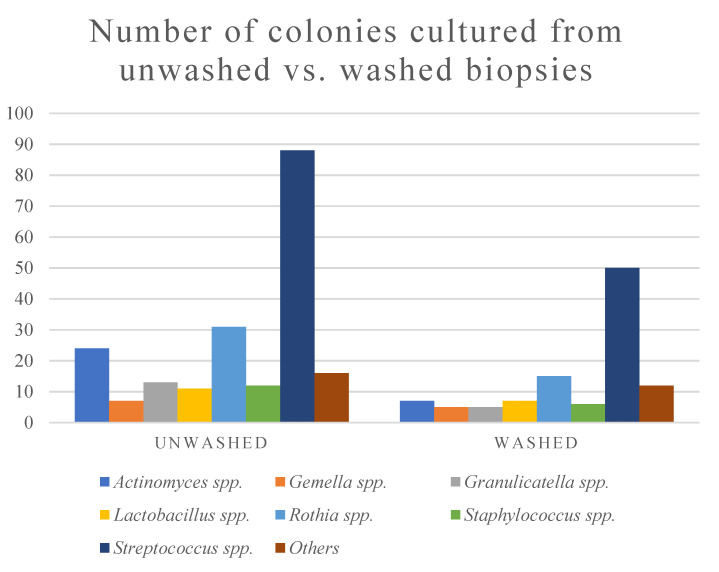
The number of bacterial species isolated from culture of gastric cancer and dyspepsia patient biopsies. “Others” include *Bacillus* spp., *Corynebacterium* spp., *Enterobacter* spp., *Enterococcus* spp., *Haemophilus* spp, *Micrococcus* spp., *Neisseria* spp., and *Stenotrophomonas* spp.

**Figure 3 jcm-09-01882-f003:**
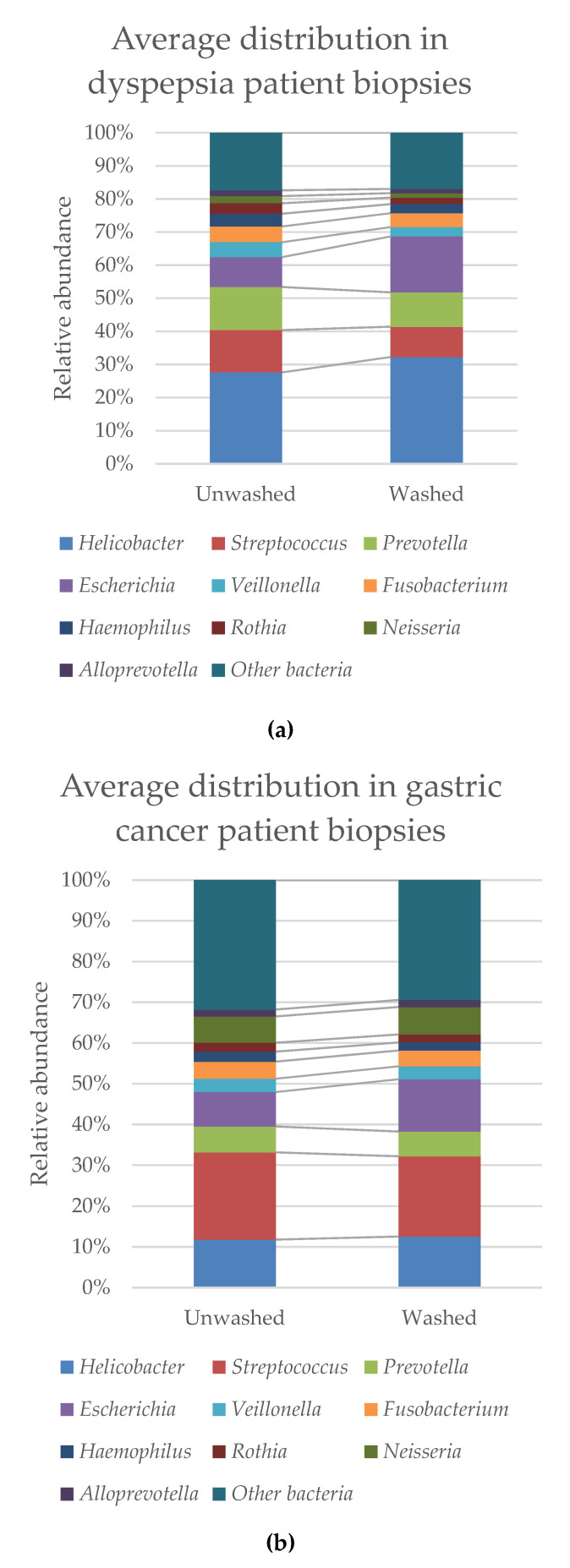
Comparison of the bacterial reads from microbiome analysis of unwashed and washed biopsies from (**a**) dyspepsia patients and (**b**) gastric cancer patients (percentages are listed in Table 4). Comparison of (**c**) the alpha-diversity shown by a Shannon Index, *p* = 0.22581 and (**d**) the beta-diversity between the two groups shown by PCoA plot, *p* = 0.801199, in unwashed biopsies (red) and washed biopsies (blue).

**Figure 4 jcm-09-01882-f004:**
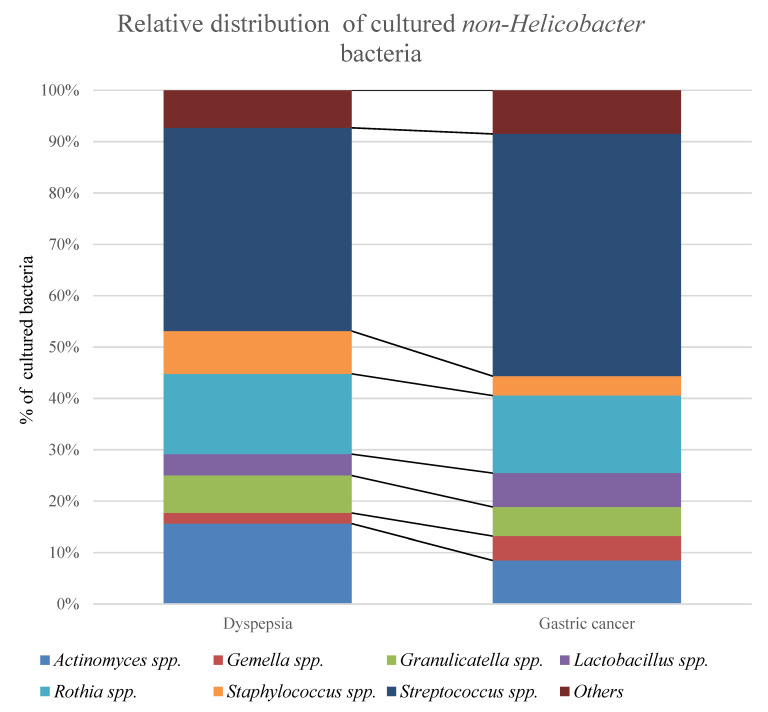
The relative distribution of non-*Helicobacter* bacteria cultured from the unwashed biopsies showed that *Streptococcus* spp. were predominant in both dyspepsia patients and gastric cancer patients. The percentages can be found in Table 3. “Others” include *Bacillus* spp., *Corynebacterium* spp., *Enterobacter* spp., *Enterococcus* spp., *Haemophilus* spp, *Micrococcus* spp., *Neisseria* spp., and *Stenotrophomonas* spp.

**Figure 5 jcm-09-01882-f005:**
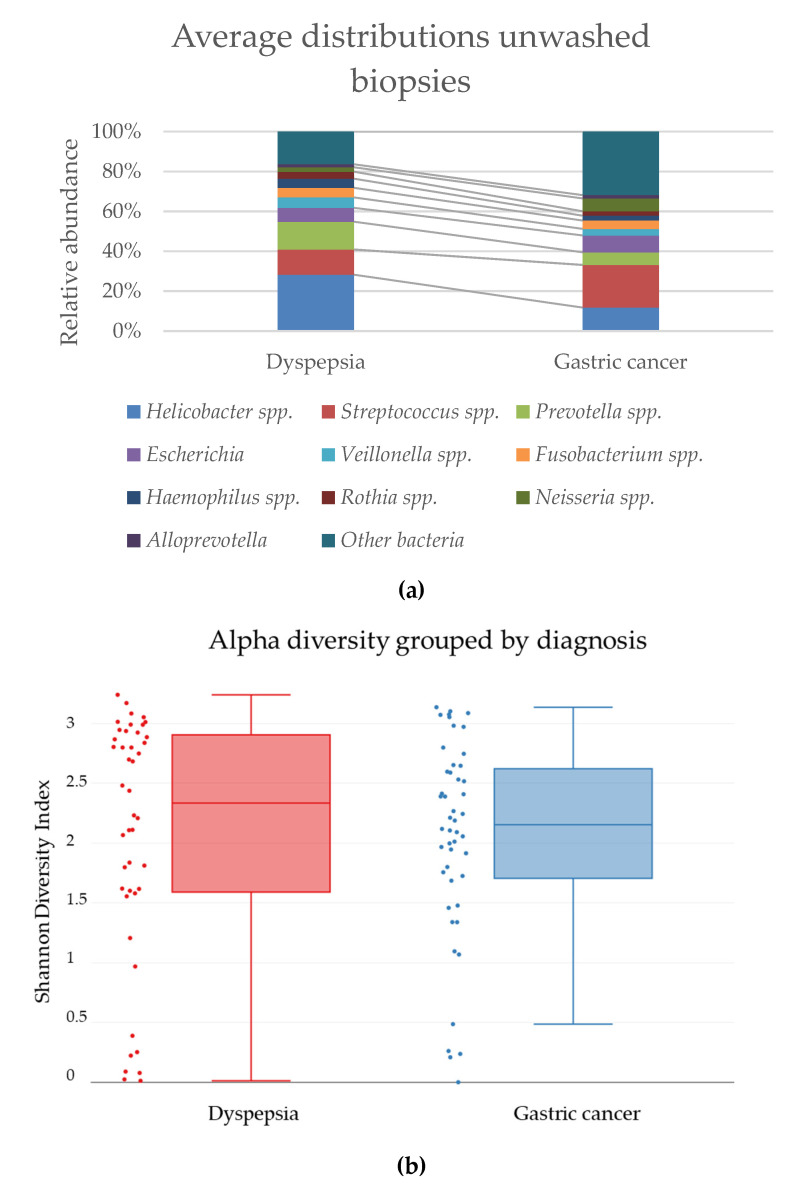
(**a**) The average relative abundance of the ten most common bacterial genera of untreated biopsies from dyspepsia patients and gastric cancer patients. Comparison of (**b**) the alpha-diversity shown by a Shannon Index, *p* = 0.556831 and (**c**) the beta-diversity between the two groups shown by PCoA plot, *p* = 0.052947, in biopsies from dyspepsia patients (red) and gastric cancer patients (blue).

**Figure 6 jcm-09-01882-f006:**
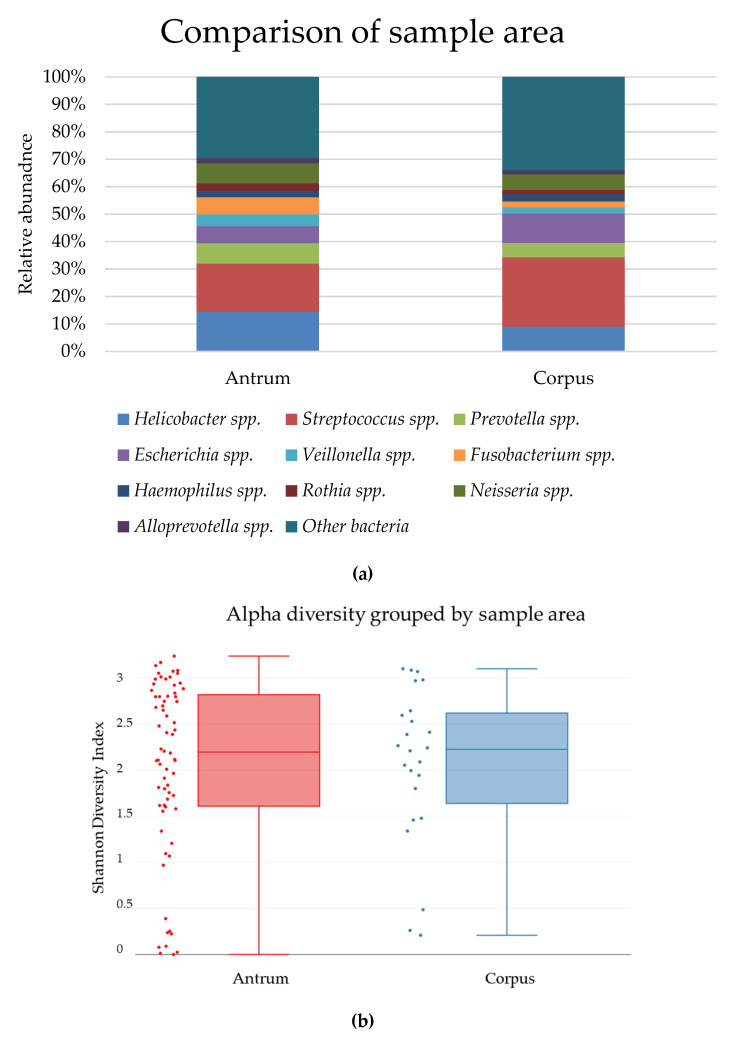
(**a**) The relative abundance of bacteria in untreated biopsies from the antrum and corpus area of gastric cancer patients. No significant differences were observed in the distributions of bacteria between the two groups. (Percentages are listed in Table 6). Comparison of (**b**) the alpha-diversity shown by a Shannon Index, *p* = 0.960995 and (**c**) the beta-diversity between shown by PCoA plot, *p* = 0.111888, in biopsies from antrum area (red) and corpus area (blue) in the stomach.

**Figure 7 jcm-09-01882-f007:**
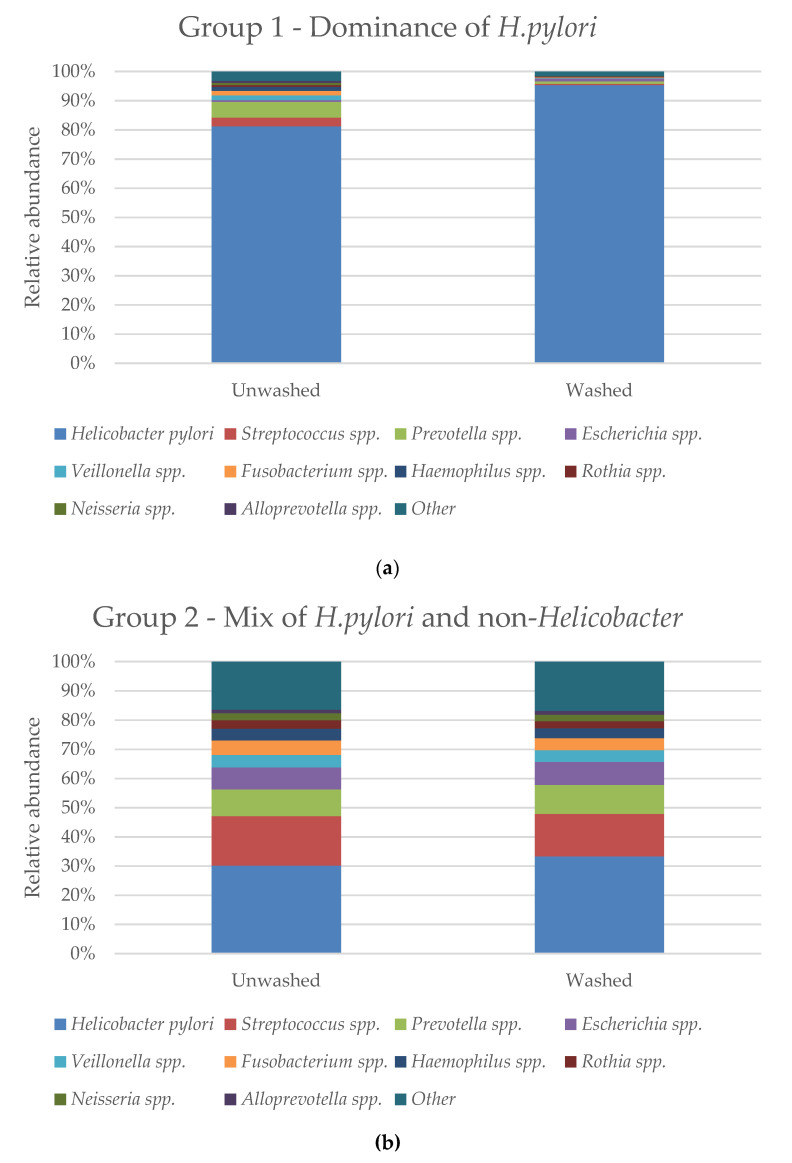
The average relative abundance divided into three groups based on their distribution of *H. pylori* and other bacteria by (**a**) dominance of *H. pylori*, (**b**) mixed presence of non-*Helicobacter* and *H. pylori*, and (**c**) absence of *H. pylori.*

**Figure 8 jcm-09-01882-f008:**
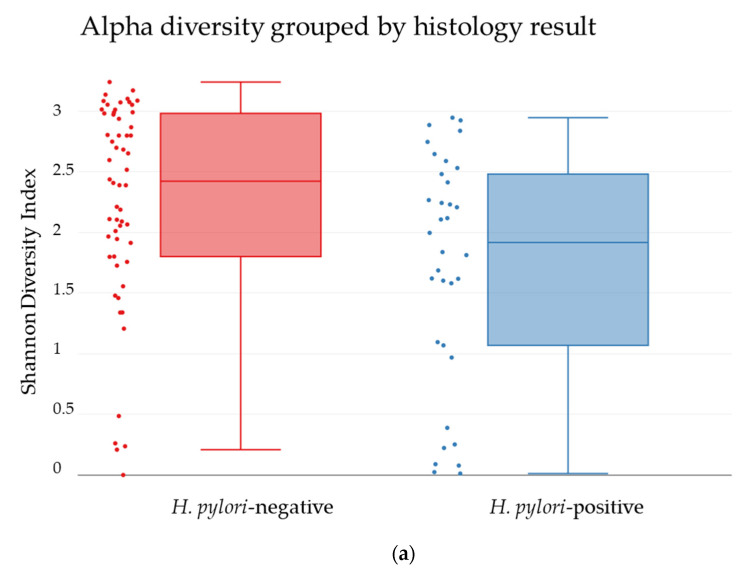
Comparison of the bacterial diversity from biopsies of patients shown negative for *H. pylori* (red) or positive for *H. pylori* (blue). (**a**) The alpha-diversity shown by a Shannon Index, *p* = 0.004353. (**b**) The beta-diversity between the two groups shown by PCoA plot, *p* = 0.000999.4.

**Table 1 jcm-09-01882-t001:** Clinical data of patients included in the study.

Diagnosis	Mean Age (years)	Gender	*H. pylori* Status (Histology)	Lauren Classification	G Stage	Proximal/Distal Part
Gastric adenocarcinoma (*n* = 12)	62.2	Female (*n* = 5)	Negative (*n* = 9)	Diffuse (*n* = 6)	2 (*n* = 5)	Proximal (*n* = 3)
		Male (*n* = 7)	Positive (*n* = 3)	Intestinal (*n* = 4)	3 (*n* = 7)	Distal (*n* = 9)
				Mixed (*n* = 2)		
Dyspepsia (*n* = 22)	48.1	Female (*n* = 18)	Negative (*n* = 11)	NA	NA	NA
		Male (*n* = 4)	Positive (*n* = 11)			

NA: Not applicable.

**Table 2 jcm-09-01882-t002:** Change in growth of bacterial species cultured from the biopsy pairs.

Growth Result	Dyspepsia Patients	Gastric Cancer Patients
No bacterial growth in either biopsy	8 biopsy pairs	3 biopsy pairs
Growth in unwashed but not in washed biopsy	4 biopsy pairs	5 biopsy pairs
Reduced growth in the washed biopsy	9 biopsy pairs	9 biopsy pairs
Unchanged growth	1 biopsy pairs	4 biopsy pairs
Increased growth in the washed biopsy	0 biopsy pairs	3 biopsy pairs

**Table 3 jcm-09-01882-t003:** Distribution of total cultured bacterial groups.

	Dyspepsia		Gastric Cancer	
	Unwashed	Washed	Unwashed	Washed
*Streptococcus* spp.	40%	43%	47%	49%
*Rothia* spp.	16%	14%	15%	14%
*Actinomyces* spp.	16%	9%	9%	6%
*Staphylococcus* spp.	8%	6%	4%	6%
*Granulicatella* spp.	7%	6%	6%	4%
*Lactobacillus* spp.	4%	3%	7%	8%
*Gemella* spp.	2%	9%	5%	3%
*Enterococcus* spp.	1%	3%	6%	8%
*Micrococcus* spp.	2%	3%	2%	0%
*Corynebacterium* spp.	2%	3%	-	-
*Stenotrophomonas* spp.	1%	0%	-	-
*Neisseria* spp.	1%	0%	-	-
*Enterobacter* spp.	0%	3%	-	-
*Bacillus* spp.	-	-	1%	0%
*Haemophilus* spp.	-	-	0%	3%

**Table 4 jcm-09-01882-t004:** Mean and standard error of the 10 most common bacterial groups as percentage of total bacterial reads in the microbiome analysis.

	Dyspepsia Biopsiy Pairs (*n* = 22)		Gastric Cancer Biopsy Pairs (*n* = 24)	
	Unwashed (% ± SEM)	Washed (% ± SEM)	Unwashed (% ± SEM)	Washed (% ± SEM)
*Helicobacter* spp.	28.3 ± 7.9	31.2 ± 8.4	11.8 ± 4.4	12.7 ± 5.0
*Streptococcus* spp.	12.6 ± 1.8	9.6 ± 1.8	21.3 ± 4.3	20.1 ± 4.3
*Prevotella* spp.	13.9 ± 2.7	11.0 ± 2.5	6.4 ± 1.1	6.1 ± 1.3
*Escherichia* spp.	7.0 ± 2.9	10.4 ± 3.8	9.5 ± 3.6	11.9 ± 3.6
*Veillonella* spp.	5.3 ± 0.9	3.4 ± 0.9	3.5 ± 1.0	3.7 ± 0.7
*Fusobacterium* spp.	4.8 ± 1.0	4.3 ± 1.5	4.2 ± 1.4	3.7 ± 1.2
*Haemophilus* spp.	4.6 ± 1.3	3.2 ± 0.9	2.4 ± 0.5	2.0 ± 0.5
*Rothia* spp.	3.5 ± 0.8	1.9 ± 0.6	2.2 ± 0.4	2.0 ± 0.7
*Neisseria* spp.	2.2 ± 0.7	1.5 ± 0.6	6.5 ± 2.7	6.8 ± 2.5
*Alloprevotella* spp.	1.5 ± 0.5	1.4 ± 0.4	1.7 ± 0.7	1.8 ± 0.7
Other bacteria	17.4 ± 2.7	17.0 ± 3.1	30.57 ± 5.8	29.3 ± 5.2

**Table 5 jcm-09-01882-t005:** Mean and standard error of the most common bacterial groups belonging to “other bacteria” as percentage of total bacterial reads in the microbiome analysis.

	Dyspepsia Biopsy Pairs (*n* = 22)		Gastric Cancer Biopsy Pairs (*n* = 24)	
	Unwashed (% ± SEM)	Washed (% ± SEM)	Unwashed (% ± SEM)	Washed (% ± SEM)
*Abiotrophia* spp.	0.013 ± 0.007	0.007 ± 0.009	0.104 ± 0.055	0.128 ± 0.047
*Aggregatibacter* spp.	0.139 ± 0.043	0.134 ± 0.044	0.103 ± 0.038	0.187 ± 0.086
*Atopobium* spp.	0.359 ± 0.113	0.31 ± 0.128	0.328 ± 0.064	0.335 ± 0.086
*Campylobacter* spp.	0.507 ± 0.141	0.452 ± 0.104	0.423 ± 0.121	0.314 ± 0.116
*Capnocytophaga* spp.	0.455 ± 0.19	0.308 ± 0.123	0.329 ± 0.087	0.274 ± 0.079
*Catonella* spp.	0.311 ± 0.098	0.212 ± 0.056	0.093 ± 0.027	0.119 ± 0.061
*Corynebacterium* spp.	0.214 ± 0.112	0.257 ± 0.175	0.047 ± 0.017	0.08 ± 0.033
*Dialister* spp.	0.304 ± 0.089	0.199 ± 0.083	0.31 ± 0.132	0.242 ± 0.101
*Eubacterium* spp.	0.123 ± 0.03	0.181 ± 0.059	0.186 ± 0.07	0.277 ± 0.147
*Filifactor* spp.	0.172 ± 0.076	0.262 ± 0.139	0.043 ± 0.012	0.08 ± 0.044
*Flavobcaterium* spp.	1.235 ± 0.975	1.27 ± 0.927	0.164 ± 0.063	0.575 ± 0.375
*Gemella* spp.	0.827 ± 0.228	0.82 ± 0.313	0.861 ± 0.211	1.446 ± 0.433
*Granulicatella* spp.	1.046 ± 0.219	1.093 ± 0.206	1.333 ± 0.405	1.603 ± 0.431
*Lachnoanaerobaculum* spp.	0.296 ± 0.066	0.271 ± 0.106	0.457 ± 0.156	0.435 ± 0.137
*Lactobacillus* spp.	0.013 ± 0.007	0.012 ± 0.006	1.925 ± 1.245	10.5 ± 8.878
*Leptotrichia* spp.	0.459 ± 0.196	0.967 ± 0.516	0.728 ± 0.362	0.42 ± 0.177
*Megasphaera* spp.	0.554 ± 0.188	0.448 ± 0.143	0.219 ± 0.062	0.196 ± 0.059
*Oribacterium* spp.	0.432 ± 0.09	0.336 ± 0.11	0.329 ± 0.113	0.632 ± 0.327
*Parvimonas* spp.	0.229 ± 0.076	0.31 ± 0.188	1.478 ± 0.833	2.508 ± 1.555
*Peptostreptococcus* spp.	0.135 ± 0.045	0.13 ± 0.066	2.517 ± 1.448	3.319 ± 1.892
*Porphyromonas* spp.	1.765 ± 0.52	1.607 ± 0.653	0.666 ± 0.198	1.241 ± 0.566
*Propionibacterium* spp.	0.058 ± 0.031	0.082 ± 0.026	0.141 ± 0.056	0.256 ± 0.107
*Selenomonas* spp.	0.167 ± 0.059	0.231 ± 0.082	0.087 ± 0.022	0.088 ± 0.031
*Solobacterium* spp.	0.281 ± 0.062	0.282 ± 0.103	0.674 ± 0.29	0.603 ± 0.217
*Staphylococcus* spp.	0.403 ± 0.168	0.627 ± 0.153	0.292 ± 0.077	0.728 ± 0.351
*Stenotrophomonas* spp.	0.679 ± 0.54	1.128 ± 0.769	0.156 ± 0.052	0.495 ± 0.325
*Stomatobaculum* spp.	0.328 ± 0.12	0.365 ± 0.115	0.561 ± 0.332	0.775 ± 0.483
*Treponema* spp.	0.211 ± 0.076	0.391 ± 0.218	0.067 ± 0.031	0.056 ± 0.019

**Table 6 jcm-09-01882-t006:** Mean and standard error as percentage of the 10 most common bacteria as percentage of total bacterial reads in the microbiome analysis of biopsies from gastric cancer patients (*n* = 12).

	Antrum Area (% ± SEM)	Cancer Area (% ± SEM)
*Helicobacter* spp.	14.5 ± 7.3	9.1 ± 4.9
*Streptococcus* spp.	17.6 ± 5.1	25.1 ± 6.9
*Prevotella* spp.	7.4 ± 1.6	5.3 ± 1.5
*Escherichia* spp.	6.1 ± 2.7	12.9 ± 6.6
*Veillonella* spp.	4.3 ± 1.8	2.7 ± 2.0
*Fusobacterium* spp.	6.2 ± 2.6	2.1 ± 0.8
*Haemophilus* spp.	2.2 ± 0.7	2.6 ± 0.9
*Rothia* spp.	2.8 ± 0.8	1.6 ± 0.5
*Neisseria* spp.	7.3 ± 4.4	5.6 ± 3.2
*Alloprevotella* spp.	1.9 ± 1.0	1.5 ± 0.9
Other bacteria	29.6 ± 7.6	34.1 ± 8.8

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
