# Peer review of "Transient and Persistent Gastric Microbiome: Adherence of Bacteria in Gastric Cancer and Dyspeptic Patient Biopsies after Washing"

_jcm, 2020, doi:10.3390/jcm9061882_

Round 1
Reviewer 1 Report
The purpose of this study, which is to evaluate the transient and persistent bacterial communities of gastric biopsies, is interesting, and then the significant results were discussed.
It is important the association of detected non-Helicobacter pylori on the gastric disorder, especially gastric cancer.
(1) Does the contamination of transient bacteria always occur? or is this contamination induced by biopsies? If the contamination of transient bacteria is constantly occurring in the stomach, are these transient bacteria involved in gastric disorders?
(2) Can the author explain the relationship between detected commensal bacteria in the cancer area and gastric carcinogenesis by referring to previous studies? And are washed (or not washed) gastric biopsies an effective way to reveal the involvement of the gastric microbiota in the development of gastric cancer?
Author Response
Dear reviewer,
1) We describe that the stomach is exposed to oral commensal bacteria when eating or swallowing, which is what we observe as contamination. In the Introduction we mention that H. pylori is the only bacterium involved in pathogenesis of the healthy stomach. However, we discuss the possibility for other bacteria to exploit a diseased stomach to further promote malignancies. It is a very interesting point when considering pathogenesis.
2) In section 1.4 previous studies of the microbiota in gastric cancer patients are described. Line 570-571 describes further studies in this area, which may also be used to investigate the involvement of the gastric microbiota on immune activation and cancer development.
Reviewer 2 Report
In this study, the authors described the transient and persistent gastric microbiota with a further focus on gastric cancer and dyspeptic patients. The introduction is exhaustive and explicative, the aim of the study is clear and the enrollment of the patients is appropriate. However, there are some inconsistences between the data performed with the culturing method and the 16S rRNA gene amplicon sequence, that in my opinion, make the conclusions difficult to interpret. I suggest to the authors to revise their data again and compact the most important findings making the result sections more consistent and moreover improve the data visualization. In addition, I have further concerns reported below.
Introduction:
Lines 73-75: The authors need to specify the family or the genera found by Maldonado-Contreras et al. The phylum level is too general and it’s important to mention more refined taxonomic levels. Moreover, a comparison with the more known bacteria taxa of the gut (colon) microbiota would be an important point to mention in the discussion.
Lines 88-89: The authors should mention which taxa compose the core microbiome described by Nardone and Compare.
Lines 152-153:
The authors mentioned in the introduction that the stomach microbiota is composed by a core microbiota .Why did the authors hypothesize the presence of only H.pylori when the colonization of other species as Lactobacillus and Streptococcus is described in a healthy stomach? (Engstrand and Lindberg, 2013).
Methods:
-I didn’t find in the methods an accession number indicating the sequence data submission. Did the author submit their sequence data in one of the available databases? If yes the author need the write the accession number in the method section.
-Use the same terminology in all the paper: use the term 16S rRNA gene-targeted amplicon sequencing in all the text.
It is already described in the methods that the DNA was extracted, moreover 16S rRNA is more common used as terminology.
Results:
The results seem inconsistent between the cultivation method and 16S rRNA gene amplicon sequence. The format of the graphs would be more visually appealing using the same style, for example using GraphPad or R as already mentioned in the methods. Moreover, the results section needs to be less redundant. I would suggest to combined more figures together. Here my comments regarding this part:
-Table 2: The table would be more informative if the authors would specify from which patients the biopsies come from.
-Figure 1: A box plot or a bar chart is more appropriate to compare the 2 groups.
-Table 3: Was there a significant difference between dyspepsia and gastric cancer groups in term of numbers of colonies detected? Looking into table 3 , it doesn’t seem statistically different, but the authors should report that they made a statistical test to search for significance between the 2 groups.
- Why Helicobacter spp. was found using the 16S amplicon sequencing but not using the culturing method?
The authors needs to make some comment on this regard.
-Moreover, the most abundant genera detected with the 2 methods don’t match. According to the bar charts, only two genera are shared (comparison between figure 1 and 2).
The authors need to revise these data and explain why there is be such a big difference.
-Figure 2, Figure 5, Figure 7: the “% of total bacterial reads” means ”relative abundance “? If yes the authors need to the change the axis name otherwise they need to remade the bar chart using the relative abundances.
Moreover, in the legend the authors should specify which bacteria are included in the “other” group.
-Figure 3b and Figure 6b: Was the PCoA performed at phylum level? It’s more appropriate to use the OTUs or the ASVs or at least the genera level.
-Figure 4: were the taxa comparisons between the two groups significant?
-Lines 288-311. The authors described that Lactobacillus spp. was isolated from several biopsies from cancer patients, but in the microbiome analysis (16S rRNA gene-targeted amplicon sequencing) no lactobacillus was found?
The most abundant genera detected with the 2 methods don’t match as in Figure 1 and 2. Only two genera are shared (comparison between figure 4 and 5). The same comment is valid for the discussion lines 391-401 where the authors made a conclusion based only on the cultivation method.
-Lines 302-304. The authors mention that the average relative abundance of H. pylori was not significantly different between untreated dyspepsia patients and gastric cancer patient biopsies but the title of the paragraph make thinking of the opposite. Looking Figure 5, there is a visible difference in the relative abundances between the two groups concerning H.pylori, but since the difference is not significant the authors need to change the title of the paragraph at line 302.
-Figure 2 and 3, Figure 5 and 6, Figure 7 and 8 can be combined together as figure panels for a better graphical visualization and clarity.
-Line 351: The authors mentioned that H.pylori was detected by histology. An histological images would be appropriate in this paragraph. Moreover, the histological procedure need to be included in the methods.
Discussion
-The author should comment of the presence of Staphylococcus spp. in the stomach microbiota.
-line 386-389: Here the authors concluded that there are other persistent bacteria in the stomach rather than only H.pylori.
Moreover, the author should present also the 16S data of their negative controls in order to be able to conclude which bacteria were really in the biopsies. Biopsies are low biomass samples, therefore the DNA extracted is often in low concentration. For this type of samples is necessary to include negative extraction controls and PCR water controls as the authors reported in the methods. I suggest to explained which negative controls were collected and these data need to be shown in the results.
- lines 391-415: The authors based here their conclusions only on the cultivation method. The data are good discussed but the lactobacillus were not detected in the 16S rRNA gene-targeted amplicon sequencing. The data coming from the cultivation method and the 16S rRNA sequencing data show that only 2 genera are shared. The authors need to comment more on it.
-Line 477-478: The conclusive sentence need to be better explained, for example specifying the “further large-scale studies” in a more clear and detailed way.
-line 452-453: “Culture-independent methods such as 16S rDNA gene sequencing or microbiome analysis may provide detailed information about the bacterial composition, however detection of DNA does not differentiate between live and dead bacteria, and between residents and contamination”
The authors have already performed the microbiota analysis in this study, what the sentence is referring to?
By using negative controls is possible to detect contaminations. See this publication: Salter S.J, Cox M.J, Turek E.M, Calus S.T, Cookson W.O, Moffatt M.F, et al. Reagent and laboratory contamination can critically impact sequence-based microbiome analyses BMC Biology 12, 87 (2014).
-Line 455-456: DNAase treatment remove all the DNA and this treatment is used only for RNA extraction.
Minor:
-Line 24-25: Please change the sentence as follow:
“washed or unwashed gastric biopsies were investigated by cultivation and microbiota analysis (16S rRNA gene-targeted amplicon sequencing) for the distribution of H. pylori and other non-Helicobacter bacteria”.
-Line 51: change the word “increased “ with “higher”.
-Line 88: change the word “affected “ with “influenced”.
-Line 89: bacterial colonization is not an appropriate term in the sentence, please delete the word.
-Line 127: change the word “persons” with “ individuals”
-Line 407: correct with “in” (typing error).
Author Response
Dear reviewer,
The authors have now revised the manuscript according to your comments, and the following changes have been made:
(Line numbers are written as found in the most recent document)
Introduction
- The family or genera identified in the article by Maldonado-Contreras et al. was not specified in the reference, and thus we cannot specify it further in this manuscript, lines 76-78
- The taxa mentioned in the reference were added, lines 92-94
- The hypothesis was further explained, lines 162-164
Methods
- An accession number and database for the sequencing data was added, lines 245-246
- The terminology was changed to “16S rRNA gene amplicon sequencing” throughout the manuscript
Results
- Table 2 was edited to show from which patient groups the biopsies were from.
- The graph has been edited to a bar chart, Figure 2
- Statistics was not performed on the results from the culture, only on the result from microbiome analysis.
- It was commented why Helicobacter was identified in the microbiome analysis and not in the culture, lines 408-410 and 531-533
- The axis name was changed to “Relative abundance” in figures 3, 5, 6, and 7
- It was not specified exactly which bacteria belong in the “other bacteria” group, as more than 100 genera were included, and many were only present in few biopsies. A list of bacterial groups identified in several biopsies was added, lines 297-304
- More information about the PCoA analysis was added in the methods, line 260
- A comment on the difference in Lactobacillus identification between culture and microbiome analysis was added, lines 472-482.
- The genera identified in both methods were commented, lines 525-531
- The paragraph title and number was changed, line 344
- Figures 2 and 3, 5 and 6, 7 and 8 from the original manuscript have been combined as panels with new figure numbers Figure 3, 5 and 7
- The histological procedure and an image of a sample positive for pylori were added in the methods section, lines 178-180, Figure 1
Discussion
- The presence of Staphylococcus was commented, lines 443-447
- The line was edited to suggest further studies, line 450-452.
- The data from negative controls of the 16S rRNA gene amplicon sequencing were not included in the manuscript, as the system BION may experience issues when handling data sets from empty sample
- The “large-scale studies” were further specified, lines 570-572.
- The sentence commenting culture-independent methods was edited, lines 543-545
- The sentence was edited, and “analysis” was replaced with “DNA extraction”, line 547
Minor changes
- The sentence was changed according to the suggestion, lines 25-27
- The word “increased” was changed to “higher”, line 52
- The word “affected” was changed to “influenced”, line 93
- The term “bacterial colonization” was removed from the sentence, line 94
- The word “persons” was changed to “individuals”, line 134
- The typing error was corrected, line 471
Round 2
Reviewer 2 Report
Line 245: The authors should write a couple of sentences regarding how the BION-META analysis was performed . For examples how the reads were clustered and which sequence identity percentage was used.
Lines 297-304: Thanks for the explanation. Can the author specify also the mean relative abundance of these genera? Or at least write that the relative abundance of those taxa, when present, was lower than 1% for example.
Lines 357-358: “or between the distribution of phyla in the groups “. The PCoA was performed based BIONs species-level classification on Bray Curtis dissimilarity (line 260). The authors should substitute the word phyla with species. The comment is valid also for line 312.
Line 533: Can the author better explain the sentence: “most likely caused by a prolonged storage in a medium unsuited for low temperatures”
Lines 546-548: The sentence is incorrect. Please remove the sentence:“DNase treatment of the biopsies prior to DNA extraction analysis would remove free DNA, but might also increase the risk of false-negative samples.”
Line 570-571: Please remove “ different analysis types”.
Author Response
Dear editors and reviewer,
The authors have now revised the manuscript according to the reviewer’s comments, and the following changes have been made:
(Line numbers are written as found in the document where the changes are shown).
Comment: “Line 245: The authors should write a couple of sentences regarding how the BION-META analysis was performed . For examples how the reads were clustered and which sequence identity percentage was used.”
- Further description of the BION-META analysis was added, lines 242-245
Comment: “Lines 297-304: Thanks for the explanation. Can the author specify also the mean relative abundance of these genera? Or at least write that the relative abundance of those taxa, when present, was lower than 1% for example.”
- The genus Actinomyces, Enterococcus and Lactococcus were removed from the list, while Campylobacter, Filifactor and Stenotrophomonas were added. The mean relative abundance and standard error have been added in a Table 5.
Comment: “Lines 357-358: “or between the distribution of phyla in the groups “. The PCoA was performed based BIONs species-level classification on Bray Curtis dissimilarity (line 260). The authors should substitute the word phyla with species. The comment is valid also for line 312.”
- The word “phyla” was replaced with “species” in lines 307 and 353.
Comment: “Line 533: Can the author better explain the sentence: “most likely caused by a prolonged storage in a medium unsuited for low temperatures””
- The sentence was rephrased, line 527-529
Comment: “Lines 546-548: The sentence is incorrect. Please remove the sentence:“DNase treatment of the biopsies prior to DNA extraction analysis would remove free DNA, but might also increase the risk of false-negative samples.””
- Line 540-541 was removed
Comment: “Line 570-571: Please remove “ different analysis types”.”
- The phrase “different analysis types” was removed, line 564
In addition to the comments, Table 2, Table 6 and the figures showing data from the microbiome analysis (Figure 3ab, 5a, 6a and 7abc) were edited.
This manuscript is a resubmission of an earlier submission. The following is a list of the peer review reports and author responses from that submission.
Round 1
Reviewer 1 Report
The present study investigated the composition of gastric microbiota to evaluate the transient and persistent bacterial communities of the stomach. However, it cannot exclude the possibility of contamination by only analyzing the detection of bacterial DNA by the next-generation sequencer. In order to obtain a clear conclusion, the bacterial cells must be detected by experiments such as immunostaining analysis of the bacteria by using human biopsies tissues. There are so many papers that analyzed the gastric microbiota using human gastric biopsies that are released to clarify the relationship between gastric microbiota and gastric pathology. From these papers, although it is indicated various gastric bacterial spp as candidates to related to the disease, the relationship between the pathology and the detected bacterial species is not clearly shown. In the present study, increased Lactobacillus spp were detected in gastric cancer patients. However, it is no clearly showed the relationship between Lactobacillus spp and gastric carcinogenesis. This analysis of detecting bacterial DNA by using next-generation sequencer produces results, but there are too many unknown reasons. Since this is the experimental limitation, the author needs to more clear what novelty this paper has compared to other papers in the gastric microbiota analysis.
Reviewer 2 Report
The objective of the work by Spiegelhauer et al. was to demonstrate that the washing of biopsy samples can be used to differentiate between transient and resident gastric bacteria. This appears to be a novel contribution that if successful, would have important ramifications for the field. The authors provide a series of experiments characterizing the microbiome profiles in various biopsy samples. However, their results do not adequately address the overall hypothesis. The decision was made based on the following points:
- The main part of this study was to demonstrate washing can be used as an easy technique to decipher transient from host bacteria. For such an important part of the manuscript, very little was given about the method. How long were the washes? Was this done at 4OC? How exactly were the washes performed?
- No description was provided on the patients. Even a table with age, weight, and BMI could have been included. No rationale was provided for why patients with gastric cancer were included in this study. What did this add to the primary objective?
- There was no statistics section in the methods. Most results were trends and some lacked statistical analysis, such as Figure 1.
- The authors provide an extensive amount of descriptive characterization comparing various samples. These microbiome analyses really have no bearing on answering the initial question, or if they do, their justification is not clear as is currently written. The results really do not convince the reader that washing biopsy samples truly removes transient bacteria, leaving only resident species.
In addition, some general comments are below:
- The introduction was very long. Despite this, a minimal amount of information was provided detailing what is known about transient bacteria in the stomach. Also, no basis for their method is included: what evidence supports the use of washing as a means to decipher transient from resident species? How does the washing here differ from reports that washed samples prior to sequencing?
- There was a lot of information provided about H. pylori. However, this was not really the focus of the paper.
- Report mean with variance in the data where appropriate, such as Figure 1.
- Sections in the discussion needed a clear objective. It was also uncertain why several sections (e.g. 4.2) were included.
- The inability to culture H. pylori from biopsies that were identified as positive was discouraging. It also led to questioning how reliable and indicative the culturing results in the rest of the paper were.
Reviewer 3 Report
The manuscript presented by Spiegelhauer et al. evaluates the transient and persistent microbiota in gastric biopsies from dyspepsia and gastric cancer patients. In the manuscript, the authors are also testing a method of biopsy washing to study these two group of microbiotas. This study is important to improve our understanding of the role of microbiota in the pathogenesis of gastric diseases. The authors have designed multiple different methods to determine the transient and persistent microbiota. However, the study has yet to determine whether the method effectively and accurately differentiates the transient and persistence microbiota, which will undermine the applicability of this publication for further research. In addition, the major aim of the research is not consistently focused on throughout the manuscript. There are few major issues to be addressed before publication.
Major comment 1:
The authors attempt to wash the biopsy with PBS (twice) to remove the transient microbiota from the biopsies. However, this is in contrast with the paper published by Li XX et al. (Plos One 2009), citation number 15 in the manuscript. In the paper by Li XX, stomach biopsies were washed 3 times with increasing harsh conditions and over 90% of the bacteria was found to still attach to the stomach. It was concluded that washing does not remove bacteria from the biopsies, as described below:
“…Given that rigorous washing steps were not able to separate the microbiota from the biopsies, we hypothesize that majority of the identified bacteria are associated tightly with the stomach mucosa….”
Can the authors please describe how much bacterial has been removed using the method in their current data? Can the authors also compare their method to the one by Li XX et al and address the discrepancies?
Major comment 2:
In the results section, there are issues with the figures. First, most of the figures are without statistical analysis (except Figure 6, 7, 8, 10) or error bars, making it difficult to interpret the significance of the data. Second, most of the figures are without a figure legend to describe which samples were analyzed, the number of replicates in each group and what statistical methods were used, making it difficult to understand the figures. Third, many statements in the results are without references to figures or data. Examples are line 237-238, line 296-301; these statements are not supported with figures or data. The authors should make sure that every result statement has a supporting figure, otherwise, ‘data not shown’ should be stated. Fourth, it is not clear what the research question is for each result section. The author should attempt to consistently interpret the data based on transient versus persistent microbiota, which is the major focus of the manuscript. Example such as section 3.5 is a deviation from the aim of the study.
Major comment 3:
In the discussion section, the author may not have accurately interpreted the result. The author should keep in mind that they are developing a method to better understand the transient and persistent microbiota in the stomach, therefore, this should be discussed consistently throughout the discussion. However, section 4.2, 4.4, 4.5, 4.6 did not mention the transient/persistent microbiota, while only in section 4.7 and 5 that the authors discussed this part of their results. The authors should discuss whether the aim of the paper has been achieved or provide alternative hypothesis, troubleshoot, limitation of the study and further directions if otherwise.
Similarly, some important claim made by the authors were not clearly shown in the results, such as those in section 5 –
“oral bacteria in the biospies suggest that…, though they were able to remain in the biopsies after the washing step”.
However, the data were not easily found or analyzed in the results. It would be easier for the readers to understand the paper if each section of the discussion corresponds to each section in the results in the same order.
Minor comments:
- In the method, the section 2.2 is not clear enough for other researcher to repeat the experiments. In line 152, were the plates incubated for a further 1-3 days after the single colonies were isolated? Or was the 1-3 days included as part of the 6-day culture?
- Ethical statements for human sample collections can be included in the method.